# Role of Deep Learning in Prostate Cancer Management: Past, Present and Future Based on a Comprehensive Literature Review

**DOI:** 10.3390/jcm11133575

**Published:** 2022-06-21

**Authors:** Nithesh Naik, Theodoros Tokas, Dasharathraj K. Shetty, B.M. Zeeshan Hameed, Sarthak Shastri, Milap J. Shah, Sufyan Ibrahim, Bhavan Prasad Rai, Piotr Chłosta, Bhaskar K. Somani

**Affiliations:** 1Department of Mechanical and Industrial Engineering, Manipal Institute of Technology, Manipal Academy of Higher Education, Manipal 576104, Krnataka, India; nithesh.naik@manipal.edu; 2iTRUE (International Training and Research in Uro-Oncology and Endourology) Group, Manipal 576104, Karnataka, India; drmilapshah@gmail.com (M.J.S.); sufyan.ibrahim2@gmail.com (S.I.); urobhavan@gmail.com (B.P.R.); bhaskarsomani@yahoo.com (B.K.S.); 3Department of Urology and Andrology, General Hospital Hall i.T., Milser Str. 10, 6060 Hall in Tirol, Austria; ttokas@yahoo.com; 4Department of Humanities and Management, Manipal Institute of Technology, Manipal Academy of Higher Education, Manipal 576104, Karnataka, India; 5Department of Urology, Father Muller Medical College, Mangalore 575002, Karnataka, India; 6Department of Information and Communication Technology, Manipal Institute of Technology, Manipal Academy of Higher Education, Manipal 576104, Karnataka, India; sarthak.shastri@learner.manipal.edu; 7Robotics and Urooncology, Max Hospital and Max Institute of Cancer Care, New Delhi 110024, India; 8Kasturba Medical College, Manipal Academy of Higher Education, Manipal 576104, Karnataka, India; 9Department of Urology, Freeman Hospital, Newcastle upon Tyne NE7 7DN, UK; 10Department of Urology, Jagiellonian University in Krakow, Gołębia 24, 31-007 Kraków, Poland; piotr.chlosta@uj.edu.pl; 11Department of Urology, University Hospital Southampton NHS Trust, Southampton SO16 6YD, UK

**Keywords:** artificial intelligence, deep learning, convolutional neural network, computer-aided detection, medical imaging, Gleason grading

## Abstract

This review aims to present the applications of deep learning (DL) in prostate cancer diagnosis and treatment. Computer vision is becoming an increasingly large part of our daily lives due to advancements in technology. These advancements in computational power have allowed more extensive and more complex DL models to be trained on large datasets. Urologists have found these technologies help them in their work, and many such models have been developed to aid in the identification, treatment and surgical practices in prostate cancer. This review will present a systematic outline and summary of these deep learning models and technologies used for prostate cancer management. A literature search was carried out for English language articles over the last two decades from 2000–2021, and present in Scopus, MEDLINE, Clinicaltrials.gov, Science Direct, Web of Science and Google Scholar. A total of 224 articles were identified on the initial search. After screening, 64 articles were identified as related to applications in urology, from which 24 articles were identified to be solely related to the diagnosis and treatment of prostate cancer. The constant improvement in DL models should drive more research focusing on deep learning applications. The focus should be on improving models to the stage where they are ready to be implemented in clinical practice. Future research should prioritize developing models that can train on encrypted images, allowing increased data sharing and accessibility.

## 1. Introduction

Artificial intelligence (AI) is a broad term that incorporates machine learning (ML), in which an algorithm analyzes features in a separate dataset, based on raw input data, without being explicitly programmed, and returns a specific classification [1]. Deep learning (DL) is a subset of ML which uses multilayer artificial neural networks (ANNs) to learn hierarchical representations. Unlike classic ML algorithms such as support vector networks (SVN) and random forest (RF), DL learns features from input data without relying substantially on domain knowledge developed by engineers [2]. Deep learning uses neural networks with many layers where the first layer is the input layer connected to multiple hidden layers that are finally connected to the output layer. Neural networks use a series of algorithms to recognize hidden relationships in a data set by a process similar to the human brain. Each of the interconnected layers comprises numerous nodes, which are called perceptrons. Model perceptrons are arranged to form an interconnected network in a multi-layered perceptron. The input layer, upon receiving the input, transfers patterns obtained to the hidden layers. The hidden layers are activated based on the input parameters received. Hidden layers fine-tune the inputs received until the error is minimal, after which the values of the neurons are passed to the output layer. The activation function calculates the output value, and the neural network produces its result.

Deep learning models help in diagnosing and treating urological conditions and have proved their ability to detect prostate cancer, bladder tumors, renal cell carcinoma, along with ultrasound image analysis. A general schematic diagram of DL models can be seen in Figure 1. Deep learning models have also displayed their ability to detect the needle/trocar pressure during insertion, which is an essential aspect of laparoscopic and robotic urological surgeries.

Supervised learning and unsupervised learning are the two main approaches in AI and machine learning. The primary distinction between the two approaches is the reliance on labelled data in the first, as opposed to the latter. Though the two approaches share many similarities, they also have distinct differences. Figure 2 shows the distinction between the supervised learning and unsupervised learning approach.

What is supervised learning?

The use of labelled datasets distinguishes supervised learning from other forms of machine learning. Using these datasets, algorithms can be trained to better classify data or predict results. The model’s accuracy can be measured and improved over time using labelled inputs and outputs.

Based on data mining, supervised learning can be categorised into two types: classification and regression: (a) classification tasks rely on an algorithm to reliably assign test data to specified groups. For example: supervised learning algorithms can differentiate spam from the rest of the incoming emails. Classification methods include linear classifiers, support vector machines, decision trees, and random forests. (b) Regression is used to learn about the relationship between dependent and independent variables. Predicting numerical values based on various data points is possible with regression models. Linear regression, logistic regression, and polynomial regression are all common regression algorithms.

What is unsupervised learning?

For the analysis and clustering of unlabeled data sets, unsupervised learning makes use of machine learning methods. These algorithms, which are referred to as ‘unsupervised’, find hidden patterns in data without the aid of human intervention. Three key tasks are performed by unsupervised learning models: (a) clustering, (b) association, and (c) dimensionality reduction.

Using data mining techniques such as clustering, it is possible to create groups of unlabeled data that are similar or dissimilar. Similar data points are grouped together according to the K value in K-means clustering algorithms. This method is useful for a variety of things, including image segmentation and image compression. Another unsupervised learning technique is association, which employs a separate set of criteria to discover connections among the variables in a dataset. 

Dimensionality reduction is a learning approach used when the number of features (or dimensions) in a dataset is too large. It minimizes the quantity of data inputs while yet maintaining the integrity of the data. To enhance image quality, auto encoders often utilize this technique to remove noise from visual data before it is processed further.

Over the last decade, imaging technology has significantly improved, which has made it easier for us to apply computer vision technologies for the classification and detection of diseases [3]. With the advancements in graphics processing units (GPUs) and their computational power to perform parallel processing, computer vision processing is more accessible today. Deep learning is also being used for data management, chatbots, and other facilities that aid in medical practice. Natural language processing (NLP) practices used in finding patterns in multimodal data have been shown to increase the learning system’s accuracy of diagnosis, prediction, and performance [4]. However, identifying essential clinical elements and establishing relations has been difficult as these records are usually unordered and disorganized. Urology has been at the forefront of accepting newer technologies to achieve better patient outcomes. This comprehensive review aims to give an insight into the applications of deep learning in Urology.

## 2. Search Strategy

In October 2021, Pubmed/MEDLINE, Scopus, Clinicaltrials.gov, Science Direct, Web of Science and Google Scholar were used to undertake a review of all English language literature published in the previous two decades (2000–2021). The search technique was based on PICO (Patient Intervention Comparison Outcome) criteria, in which patients were treated with AI models (I) or classical biostatistical models (C), and the efficacy of AI models was evaluated (O) [5].

Specifically, the search was conducted by using a combination of the following terms: ‘artificial intelligence’, ‘AI’, ‘machine learning’, ‘ML’, ‘convolutional networks’, ‘CNN’, ‘deep learning’, ‘DL’, ‘magnetic resonance imaging’, ‘prostate’, ‘prostate cancer’, ‘MRI’, ‘Sorensen–Dice coefficient’, ‘DSC’, ‘area under the ROC curve’, ‘AUC’, ‘Sorensen–Dice index’, ‘SDI’ and ‘computed tomography’, ‘CT’ [6].

### 2.1. Inclusion Criteria

Articles on the application of deep learning in prostate cancer diagnosis and treatment.Full-text articles, clinical trials and meta-Analysis on outcomes of analysis in Urology using deep learning.

### 2.2. Exclusion Criteria

Animal, laboratory, or cadaveric studiesReviews, editorials, commentaries or book chapters

The literature review was carried out using the inclusion and exclusion criteria mentioned. Articles were screened based on the titles and abstracts. Articles were then selected and their entire text was analyzed. For further screening of other published literature, the references list of the selected articles was individually and manually checked.

## 3. Results

### Evidence Synthesis

A total of 224 distinct articles were discovered during the initial search. Following the initial screening, 97 articles remained, with 64 left after a second screening as related to applications of deep learning in Urology. Among these articles, 24 were identified to be solely related to prostate cancer, these abstracts satisfied our inclusion criteria and were then included in the final review. The summary of all the previous studies from the literature is shown in Table 1 and Table 2 on the diagnosis and treatment of prostate cancer, respectively. 

## 4. Discussion

### 4.1. Diagnosis of Prostate Cancer Using MRI Images

Eleven studies have evaluated the application of deep learning in diagnosing prostate cancer.

Takeuchi et al. (2019) developed a DL model to predict prostate cancer using a multilayer ANN. The model was trained on images obtained from 232 patients and validated its accuracy on images obtained from 102 patients. On a test dataset, the model achieved AUC of 0.76, thereby, suggesting that neural network achieved better results as compared to a logistic regression model. However, this accuracy needs to be improved to be implemented in clinical practice [7].

Khosravi et al. (2021) used DL models to differentiate malignant and benign tumors and high- and low-risk tumors which achieved an AUC of 0.89 and 0.78, respectively. The study concluded that new images captured did not require manual segmentation and could be implemented in clinical practice [14].

Hiremath et al. (2020) used diffusion-weighted imaging fitted with monoexponential function, ADCm, employing a deep learning architecture (U-Net) to investigate the short-term test-retest repeatability of U-Net in slice- and lesion-level identification and segmentation of clinically significant prostate cancer (csPCa: Gleason grade group > 1) (U-Net). The training dataset included 112 PCa patients who had two prostate MRI exams on the same day. Two U-Net-based CNNs were trained using this dataset. The study performed three-fold cross-validation on the training set and evaluated its performance and repeatability on testing data. The CNNs with U-Net-based architecture demonstrated an intraclass correlation coefficient (ICC) between 0.80–0.83, agreement of 66–72%, and DSC of 0.68–0.72 for a slice- and lesion-level detection. These findings lay the groundwork for DL architecture’s test-retest and repeatability in identifying and segmenting clinically relevant prostate cancer on apparent diffusion coefficient maps. [11].

To summarize, MR images are most commonly used to study the applications of DL in image-based diagnosis of prostate cancer (PCa). Though the accuracy of the DL models appears to be satisfactory, the generalizability of the results across varied demographics still needs to be tested before implementing into general clinical practice.

### 4.2. Histopathological Diagnosis of Prostate Cancer Using DL Models

Three studies have evaluated the application of deep learning in the diagnosis of prostate cancer.

Kott et al. (2019) developed a DL algorithm for histopathologic diagnosis. They also performed Gleason grading of the prostate cancer biopsies. This histopathologic diagnosis and Gleason grading process are considered lengthy and time-consuming. Using ML models, this process can be made significantly faster and more efficient. The study was performed using 85 prostate biopsies from 25 patients with further magnification of up to 20x performed. The study used a deep residual CNN model with fivefold cross-validation. The DL model achieved 91.5 and 85.4% accuracy at coarse and fine-level classification, respectively. The study concluded that the model achieved excellent performance for the diagnosis as mentioned earlier; however, it needs to be tested on a larger sample size for external validation [18].

Lucas et al. (2019) performed a study using DL models for automatic Gleason pattern classification to identify grade groups from prostate biopsies. The study used a dataset containing 96 prostate biopsies from 38 patients. The Inception-v3 convolutional neural network was trained to generate probability maps. The model has a 92% accuracy in distinguishing between non-atypical and malignant regions, with a sensitivity and specificity of 90 and 93%, respectively. The study successfully demonstrates the feasibility of training CNN models to differentiate between Gleason patterns in heterogeneous biopsies [19].

The DL models have shown promising results in the histopathological diagnosis of PCa. This can definitely be added as an adjunct tool for the histopathologists to reduce the burden in terms of time and workload. However due to lack of external validation of these models, their applicability cannot be generalized as of yet.

### 4.3. Diagnosis of Prostate Cancer Using MR Based Segmentation Techniques

Four studies have evaluated the application of DL in the diagnosis of prostate cancer.

Lai et al. (2021) developed a DL CNN model to segment prostate zones and cancer regions from MRI images. The study was performed using the PROSTATEx dataset containing MRI scans from 204 patients. A SegNet model was modified and fine-tuned to perform adequately on the dataset. The study achieved a dice similarity coefficient of 90.45% for the transition zone, 70.04% for the peripheral zone, and 52.73% for the prostate cancer region. The study concluded that automatic segmentation using a DCNN model has the potential to assist in prostate cancer diagnosis [21].

Sloun et al. (2021) performed prostate segmentation of transrectal ultrasound using the DL model on MRI images. The study used three datasets with MRI-transrectal ultrasound images collected at different institutions. The study trained a U-Net model on the dataset of 436 images and achieved a median accuracy of 98%. While performing zonal segmentation, the model achieved a pixel-wise accuracy of 97 and 98% for the peripheral and transition images. The model can also self-assess its segmentation, allowing it to identify incorrect segmentations swiftly. The process of performing manual segmentation of prostate MRI images places a burden on clinicians. The authors concluded that using DL models can allow for fast and accurate segmentation of MRI images from different scanners [22].

Schelb et al. (2020) produced a comparison of prostate MRI lesion segmentation between a DL model and multiple radiologists. The study was performed using MRI images collected from 165 patients suspected to have prostate cancer. The study used U-Net models trained on the dataset of MRI images to perform segmentation. The mean Dice coefficient for manual segmentation was between 0.48–0.52, while the U-Net segmentations exhibited a Dice coefficient of 0.22. The authors concluded that smaller segmentation sizes could explain the lower Dice coefficients of the U-Net model. They also discuss how the overlapping lesions between multiple rates can be used as a secondary measure for segmentation quality in future studies [23].

Soerensen et al. (2021) performed a study to determine if DL improves the speed and accuracy of prostate gland segmentation from MRI images. The study used images from 905 subjects who underwent prostate MRI scans at 29 institutions. The study trained a ProGNet model on 805 cases and tested it on 100 independent and 56 external cases. The study found that the ProGNet model outperformed the U-Net model. The study also found that the ProGNet model achieved a Dice similarity coefficient of 0.93, outperforming radiology technicians, producing results at 35 s/case. The study concluded that DL models could be used for segmentation in targeted biopsy in routine urological clinical practice [24].

As proven, ProGNet model outperformed not only the U-Net model but also the radiology technicians in terms of speed and accuracy. However, it should be noted that authors have not compared the ProGNet model to trained and experienced urologists and radiologists. Furthermore, the accuracy of the model has to be tested across different MRI scanners.

### 4.4. Diagnosis of Prostate Cancer Using CT Images

Four studies have evaluated the application of DL in the diagnosis of prostate cancer and prostatectomy. Polymeri et al. (2019) used a DL algorithm to automate prostate cancer quantification on positron emission tomography–computed tomography (PET/CT) scans. The study looked at the possibility of PET/CT measurements as prognostic biomarkers. The training of the DL model was performed on CT scan images of 100 patients. In 45 patients with biopsy-proven hormone-naive prostate cancer, the DL algorithm was validated. The model was evaluated based on the Sørensen–Dice index (SDI) score. The SDI scores achieved were 0.78 and 0.79 for automatic and manual volume segmentation, respectively. The study demonstrated DL applications in quantifying PET/CT prostate gland uptake and its association with overall survival. The results obtained showed agreement between automated and manual PET/CT measurements. The DL model demonstrated that PET/CT indicators were strongly linked to overall survival [26].

Ma et al. (2017) performed automatic prostate segmentation using DL and multi-atlas fusion. A dataset of 92 prostate CT scans was used to conduct and assess the study. When compared to the radiologists’ manual segmentation, the model had a Dice similarity coefficient of 86.80%. The study concluded that the DL-based method can provide a valuable tool for automatic segmentation and aid clinical practice [28].

Not many studies have been performed to check the applications of DL models using PET/CT images to highlight their advantages in the same aspect. However, the nascent applications appear promising in terms of development of DL-based biomarker and prognostic models.

### 4.5. Robot-Assisted Treatment Practices

The study by Hung et al. evaluated the application of DL in the treatment of PCa and RARP. Hung et al. created a DL model to predict urinary continence recovery following radical prostatectomy with robotic assistance. The study was performed on images obtained from 79 patients. The study trained a DeepSurv model on the dataset and achieved a concordance index (C-index) of 0.6 and a mean absolute error (MAE) of 85.9 in predicting continence. The authors concluded that using automated performance metrics and clinicopathological data, the DeepSurv model could predict continence after the prostatectomy. According to the findings, experienced surgeons had greater continence rates at 3 and 6 months following the prostatectomy [29].

The application of automated performance metrics (APMs) and its impact on clinical outcome variables was very well highlighted in this study, underlining the evidence that surgical skills impact clinical outcomes. However, this was a single-institution study and requires external validation for the same.

### 4.6. Strengths, limitations, and Areas of Future Research

A wide variety of DL models were used to diagnose and treat prostate cancer. The review contains various implementations of DL which benefit the urologists. A summary of the various models used can be viewed in the table as shown (Table 3). One of the major drawbacks of the present study is the small dataset and lack of federated learning approach. Federated learning models can be implemented to improve the data collection and sharing process for research purposes. Increasing the sample size may improve the performance of multilayer DL models as a result of more sufficient training. If the sample size is increased, neural networks with more hidden layers and nodes can perform better, avoiding early over-fitting. An increase in the variables used for prostate cancer detection can also augment the performance of a neural network model. With advanced DL models such as the single shot detector model, it is possible to make predictions on a live video feed during treatment. The live feed DL models can also program robots to help during surgeries.

## 5. Conclusions

As per our review, the most common application of DL techniques has been in the diagnosis of prostate cancer using MR image-based segmentation techniques. Although the ProgNet model outperformed trained radiologists in prostate cancer detection, we cannot generalize these results. In conclusion, for clinical application, the DL models’ performance may still need improvement. As the performance of these models increases, they will become much more implementable, with many models surpassing human accuracy and efficiency.

## Figures and Tables

**Figure 1 jcm-11-03575-f001:**
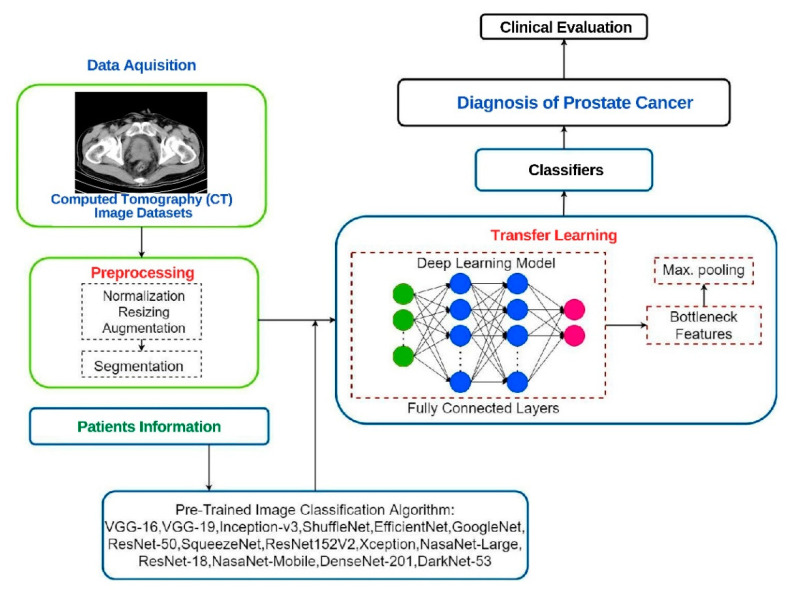
Deep learning framework for image classification.

**Figure 2 jcm-11-03575-f002:**
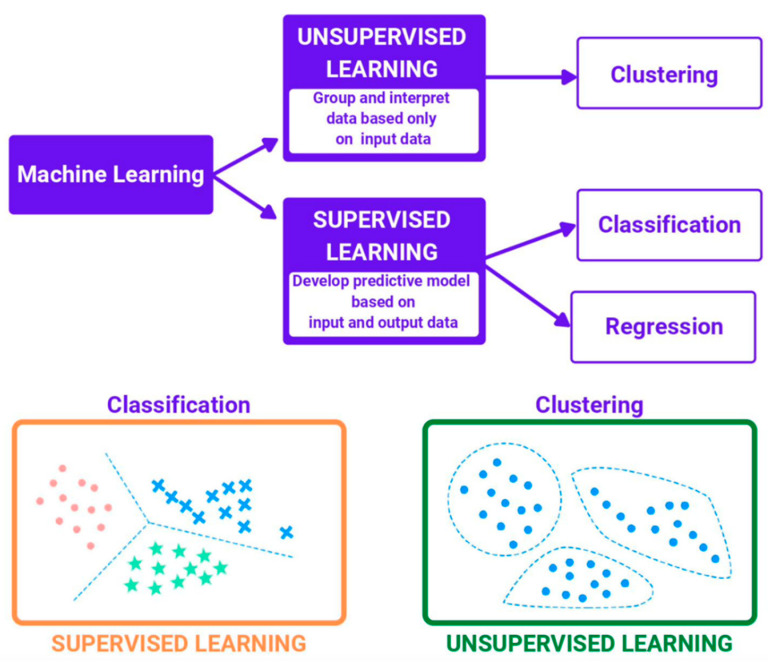
Supervised learning and unsupervised learning approach.

**Table 1 jcm-11-03575-t001:** Summary of studies on diagnosis of prostate cancer using deep learning models.

Author	Year	Objective	Sample Size (n = Patients/Images)	Study Design	Model	AUC	DSC	SDI	MAE	Sn	Sp
**A. MRI images**											
Takeuchi et al. [7]	2018	To predict PCa using DL and multilayer ANN	334 patients	Prospective	Stepwise ANN 5-hidden-layers	0.76 (Step 200)	N/A	N/A	N/A	N/A	N/A
Schelb et al. [8]	2019	To compare clinical evaluation performance with segmentation-optimized DL system in PCa diagnosis.	312 patients; T2W and diffusion images used	Retrospective	U-Net	N/A	N/A	N/A	N/A	96%	22%
Shao et al. [9]	2021	For PCa diagnosis using ProsRegNet (DL system) using MRI and histopathological data.	152 patients; T2W images and HPE slices used.	Prospective	ProsRegNet and CNNGeometric	N/A	Cohort 1: 0.979 Cohort 2: 0.971 Cohort 3: 0.976	N/A	N/A	N/A	N/A
Hiremath et al. [10]	2021	To detect csPCa using integrated nomogram using DL, PI-RADS grading and clinical factors.	592 patients; T2W and ADC images used	Retrospective	AlexNet and DenseNet	0.76	N/A	N/A	N/A	N/A	N/A
Hiremath et al. [11]	2020	To assess the test-retest repeatability of U-Net (DL system) in identification of csPCa.	112 patients; ADC/DWI images used	Prospective	U-Net	0.8	0.8	N/A	N/A	N/A	N/A
Schelb et al. [12]	2019	The use DL algorithm (U-Net) for detection, localization, and segmentation of csPCa	259 patients; T2W and DW images used.	Retrospective	U-Net	N/A	N/A	N/A	N/A	98%	24%
Yan et al. [13]	2021	For deep combination learning of multi-level features for MR prostate segmentation using a propagation DNN	80 patients; only T2W images used	Retrospective	MatConvNet	N/A	0.84	N/A	N/A	N/A	N/A
Khosravi et al. [14]	2021	To develop an AI-based model for the early detection of PCa using MR pictures tagged with histopathology information.	400 patients; T2W images used	Retrospective	GoogLenet	0.89	N/A	N/A	N/A	N/A	N/A
Shiradkar et al. [15]	2020	To find any links between T1W and T2W MR fingerprinting data and the appropriate tissue compartment ratios in PCa and prostatitis whole mount histology.	14 patients; T1W and T2 W images used	Retrospective	U-Net	0.997	N/A	N/A	N/A	N/A	N/A
Winkel et al. [16]	2020	To incorporate DL and biparametric imaging for autonomous detection and classification of PI-RADS lesions.	49 patients; T2W and DWI used	Prospective	ProstateAI	N/A	N/A	N/A	N/A	87%	50%
**B. Pathology**											
AlDubayan et al. [17]	2020	To detect germline harmful mutations in PCa using DL techniques.	1295 patients	Retrospective	Deep Variant and Genome Analysis Toolkit	0.94	N/A	N/A	N/A	CI:0.91–0.97	N/A
Kott et al. [18]	2021	To apply DL methods on biopsy specimen for histopathologic diagnosis and Gleason grading.	85 images 25 patients	Prospective	18-layer CNN	0.83	N/A	N/A	N/A	N/A	N/A
Lucas et al. [19]	2019	To determine Gleason pattern and grade group in biopsy specimen using DL	96 images 38 patients	Retrospective	Inception-v3 CNN	0.92	N/A	N/A	N/A	90%	93%

**Table 2 jcm-11-03575-t002:** Summary of studies on treatment of prostate cancer using deep learning models.

Author	Year	Objective	Sample Size	Study Design	Model	AUC	DSC	SDI	MAE	Sn	Sp
Sumitomo et al. [20]	2020	To predict risk of urinary incontinence following RARP using DL model based on MRI images	400 patients	Retrospective	CNN model	0.775	N/A	N/A	N/A	N/A	N/A
Lai et al. [21]	2021	To apply DL methods for auto-segmentation of biparametric images into prostate zones and cancer regions.	204 patients; T2W, DWI, ADC images used.	Retrospective	Segnet	0.958	N/A	N/A	N/A	N/A	N/A
Sloun et al. [22]	2020	To use DL for automated real-time prostate segmentation on TRUS pictures.	436 images 181 patients	Prospective	U-Net	0.98	N/A	N/A	N/A	N/A	N/A
Schelb et al. [23]	2020	To compare DL system and multiple radiologists agreement on prostate MRI lesion segmentation	165 patients; T2W and DWI used	Retrospective	U-Net	N/A	0.22	N/A	N/A	N/A	N/A
Soerensen et al. [24]	2021	To develop a DL model for segmenting the prostate on MRI, and apply it in clinics as part of regular MR-US fusion biopsy.	905 patients; T2W images	Prospective	ProGNet and U-Net	N/A	0.92	N/A	N/A	N/A	N/A
Nils et al. [25]	2021	To assess the effects of diverse training data on DL performance in detecting and segmenting csPCa.	1488 images; T2W and DWI images	Retrospective	U-Net	N/A	0.90	N/A	N/A	97%	90%
Polymeri et al. [26]	2019	To validate DL model for automated PCa assessment on PET/CT and evaluation of PET/CT measurements as prognostic indicators	100 patients	Retrospective	Fully CNN	N/A	N/A	0.78	N/A	N/A	N/A
Gentile et al. [27]	2021	To identify high grade PCa using a combination of different PSA molecular forms and PSA density in a DL model.	222 patients	Prospective	7-hidden-layer CNN	N/A	N/A	N/A	N/A	86%	89%
Ma et al. [28]	2017	To autonomously segment CT images using DL and multi-atlas fusion.	92 patients	NA	CNN model	N/A	0.86	N/A	N/A	N/A	N/A
Hung et al. [29]	2019	To develop a DL model to predict urinary continence recovery following RARP and then use it to evaluate the surgeon’s past medical results.	79 patients	Prospective	DeepSurv	N/A	N/A	N/A	85.9	N/A	N/A

**Table 3 jcm-11-03575-t003:** Summary of common deep learning models used in PCa management.

Diagnosis Using MRI Images	Diagnosis Using CT Images	Treatment Using MRI Images	Treatment Using CT Images
DenseNet	NiftyNet	SegNet	7-Hidden Layer CNN
U-Net	InceptionV3		U-Net
AlexNet	Stepwise Neural Network with five hidden layers	U-Net	ProgNet
MatConvNet	18-layer CNN		

## Data Availability

Not applicable.

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
