# Peer review of "Role of Deep Learning in Prostate Cancer Management: Past, Present and Future Based on a Comprehensive Literature Review"

_jcm, 2022, doi:10.3390/jcm11133575_

Round 1
Reviewer 1 Report
Comment to the authors
The authors systematically and comprehensively reviewed the current applications of deep learning models in prostate cancer diagnosis and treatment. The authors found DL models have been constantly improved and stated DL models which can be trained on encrypted images are future directions. I have some queries and comments as below.
Search strategy
- To let readers know how articles were screened and selected, could you please add a figure conforming to PRISMA flow diagrams?
Tables
- The tables included author’s name; however, did not include publication year for each study. If the authors could include publication year data in the tables, these data are very useful for readers to know how DL performance improved in the last two decade.
- The authors filled “Study Design” column with patient/image number. I think the column title and the data is not matched. Study design data should be prospective, retrospective, single centric, multi centric, etc. Patient/image number data should be included with appropriate column title (i.e. “Patients/images (n)”).
- The authors showed Accuracy/Sensitivity/Specificity of DL models in table 1. Could you please clarify that these performances are for any prostate cancer (Gleason grade group 1-5) or for clinically significant cancer (grade group >1)? In addition, specifically which type of performance (per lesion, per image slice, or per patient results) were these?
- Several measurement results (i.e. AUC, DSC, SDI, or MAE) were filled in “Accuracy” column. In binary classification, “Accuracy” is calculated by dividing true positive and true negative by total counts. Therefore, “Accuracy” column in the tables is confusing. Could you please consider preparing other column such as “Other measurement” for AUC, DSC, SDI, or MAE?
- For the studies regarding MRI, It may be useful to include the imaging modality data (T2WI, DWI, ADC, or combination etc.) used for DL.
- The layouts of Table 1 and 2 are skewed, so I felt difficulty to read the entire tables. Could you please re-check the layout of Tables?
Results
- The section titled “4.3. Diagnosis of Prostate Cancer using MRI Images” (line 120) was located after the section named “3.1. Evidence Synthesis” (line 109). Between these sections, Table 1 and 2 are located as skewed state. I suppose there might be some skipped sections.
- The section “4.4” was titled as “Diagnosis of Prostate Cancer using CT Images:”. However, the authors described the pathological diagnostic performance of DL models in the section. The section title and its contents seemed to be mismatched.
- The section “4.5” was titled as “Treatment of Prostate Cancer using MRI Images”. However, the authors mainly described the zonal/lesion segmentation performances of DL models in the section, instead of describing how DL contribute to the “treatment” of PCa using MRI. I could not agree to use “Treatment of Prostate Cancer using MRI Images” for the section title.
- The section “4.6” was titled as “Treatment of Prostate Cancer using CT Images”. The authors mainly focused on prognostic performance and zonal segmentation performance of DL using CT images. In addition, the authors referred a study by Soerensen et al. in this section (line 207). However, this study focused on DL which segment MRI images instead of CT images. I could not understand why this study was referred in this section.
- In the section “4.8. Strengths, limitations, and areas of future research”, although AUC is a popular measurement method, why did the authors mention the AUC score?
- Just referring the previous studies makes it difficult for readers to understand what the important findings are. If possible, could you please add a few sentences summarizing the findings by previous studies and the authors’ evaluation in each section?
Conclusions
- Since the conclusions were too long and complicated, I could not understand what the authors wanted to emphasize the most. I suggest keeping conclusions simple and clear. Most of sentences should be shifted to “Discussions” with references.
Minor
- Abbreviations (i.e. NLP, ICC ) should be defined at first use.
Author Response
The authors systematically and comprehensively reviewed the current applications of deep learning models in prostate cancer diagnosis and treatment. The authors found that DL models have been constantly improved and that DL models that can be trained on encrypted images are future directions. I have some queries and comments as below.
- To let readers, know how articles were screened and selected, could you please add a figure conforming to PRISMA flow diagrams?
Response: Thank you for the genuine query and comment. This was a narrative review, and although we adhered closely to PRISMA guidelines, our search strategy did not strictly adhere to the PRISMA guidelines so we do not have the PRISMA flowchart.
Tables
- The tables included the author’s name; however, did not include the publication year for each study. If the authors could include publication year data in the tables, these data are very useful for readers to know how DL performance improved in the last two decades.
Response: This is very thoughtful keeping in mind the usefulness and ease of the reader. As per advice, we have added the year of publication as a separate column in the table.
- The authors filled the “Study Design” column with the patient/image number. I think the column title and the data are not matched. Study design data should be prospective, retrospective, single centric, multi centric, etc. Patient/image number data should be included with the appropriate column title (i.e. “Patients/images (n)”).
Response: Your advice is duly noted. The necessary correction has been made to the tables. The sample size column has been added and the content in the study design column has been appropriately changed.
- The authors showed the Accuracy/Sensitivity/Specificity of DL models in table 1. Could you please clarify that these performances are for any prostate cancer (Gleason grade group 1-5) or for clinically significant cancer (grade group >1)? In addition, specifically which type of performance (per lesion, per image slice, or per patient results) were these?
Response: Thank you for the suggestion. However, there is no clear mention in most of the studies regarding the specific grade group and performance of the model per slide/ lesion/ slice or patient. This can however be studied in-depth as a separate topic altogether. However, if the reviewer or editor insists on including this, we will be happy to do so although there will be lots of gaps in the table.
- Several measurement results (i.e. AUC, DSC, SDI, or MAE) were filled in the “Accuracy” column. In binary classification, “Accuracy” is calculated by dividing true positive and true negative by total counts. Therefore, the “Accuracy” column in the tables is confusing. Could you please consider preparing other columns such as “Other measurement” for AUC, DSC, SDI, or MAE?
Response: Thank you for the advice. We have added the necessary columns of AUC, DSC, SDI and MAE along with the existing columns of sensitivity and specificity.
- For the studies regarding MRI, It may be useful to include the imaging modality data (T2WI, DWI, ADC, or combination, etc.) used for DL.
Response: Thank you once again for your advice. The necessary changes have been made and added to the table.
- The layouts of Tables 1 and 2 are skewed, so I felt difficulty reading the entire tables. Could you please re-check the layout of the Tables?
Response: Sorry for the error. We have reframed the layout of the table for convenient reading.
Results
- The section titled “4.3. Diagnosis of Prostate Cancer using MRI Images” (line 120) was located after the section named “3.1. Evidence Synthesis” (line 109). Between these sections, Tables 1 and 2 are located as skewed states. I suppose there might be some skipped sections.
Response: Thank you for correctly pointing out our error. We have rectified the error and made the necessary changes.
- The section “4.4” was titled as “Diagnosis of Prostate Cancer using CT Images:”. However, the authors described the pathological diagnostic performance of DL models in the section. The section title and its contents seemed to be mismatched.
Response: Thank you once again for pointing out the mistake. Sorry for the error. We have correctly changed the title to “Histopathological Diagnosis of Prostate Cancer using DL models”.
- Section “4.5” was titled as “Treatment of Prostate Cancer using MRI Images”. However, the authors mainly described the zonal/lesion segmentation performances of DL models in the section, instead of describing how DL contributes to the “treatment” of PCa using MRI. I could not agree to use “Treatment of Prostate Cancer using MRI Images” for the section title.
Response: We appreciate your advice. We have made the necessary changes to the title. It has been changed to “Diagnosis of Prostate cancer using MR-based segmentation techniques”.
4.3. Diagnosis of Prostate cancer using MR based segmentation techniques
4.4. Diagnosis of prostate cancer using CT images
4.5. Robot-assisted treatment practices
- Section “4.6” was titled as “Treatment of Prostate Cancer using CT Images”. The authors mainly focused on prognostic performance and zonal segmentation performance of DL using CT images. In addition, the authors referred to a study by Soerensen et al. in this section (line 207). However, this study focused on DL which segment MRI images instead of CT images. I could not understand why this study was referred to in this section.
Response: Thank you for pointing out our mistake. We have made the necessary changes in the manuscript.
- In the section “4.8. Strengths, limitations, and areas of future research”, although AUC is a popular measurement method, why did the authors mention the AUC score?
Response: We have made the changes in the mentioned paragraph and removed the AUC score from the discussion.
- Just referring to the previous studies makes it difficult for readers to understand what the important findings are. If possible, could you please add a few sentences summarizing the findings of previous studies and the authors’ evaluation in each section?
Response: The following are the changes added to the manuscript:
Section 4.1: To summarize, MR images are most commonly used to study applications of DL in the image-based diagnosis of Prostate Cancer (PCa). Though the accuracy of the DL models appears to be satisfactory, the generalizability of the results across varied demographics still needs to be tested before implementing into general clinical practice.
Section 4.2: The DL models have shown promising results in the histopathological diagnosis of PCa. This can definitely add as an adjunct tool for histopathologists to reduce the burden in terms of time and workload. However, due to the lack of external validation of these models, their applicability cannot be generalized as of yet.
Section 4.3: As proven, the ProGNet model outperformed not only the U-Net model but also the radiology technicians in terms of speed and accuracy. However, it should be noted that the authors have not compared the ProGNet model to trained and experienced urologists and radiologists. Also, the accuracy of the model has to be tested across different MRI scanners.
Section 4.4: Not many studies have been performed to check the applications of DL models using PET- CT images to highlight their advantages in the same aspect. However, the nascent applications appear promising in terms of the development of DL-based biomarkers and prognostic models.
Section 4.5: The applications of Automated Performance Metrics (APMs) and its impact on clinical outcome variable was very well highlighted in this study reassuring the evidence that surgical skills impact clinical outcomes. However, this was a single-institution study and requires external validation for the same.
Conclusions
- Since the conclusions were too long and complicated, I could not understand what the authors wanted to emphasize the most. I suggest keeping conclusions simple and clear. Most the sentences should be shifted to “Discussions” with references.
Response: Your advice is duly noted and the necessary changes are made in the manuscript.
Minor
- Abbreviations (i.e. NLP, ICC ) should be defined at first use.
Response: Duly noted and necessary changes made and marked. NLP is Natural language processing and ICC is the intraclass correlation coefficient.

Reviewer 2 Report
In the manuscript “Role of Deep Learning in Prostate Cancer management: Past, present and future based on a comprehensive literature review” by Nithesh et al, the authors have done a study on the present applications of Deep Learning (DL) in prostate cancer diagnosis and treatment. It’s a good job for which has certain suggestive significance for the rapid development of artificial intelligence in urinary tumor. I have several suggestions to improve the manuscript:
1.Why the database here not used in your study? PsycINFO, Econlit, CINAHL, EMBASE
2.I would suggest you to display article selection by Preferred Reporting Items for Systematic Reviews and Meta-Analyses (PRISMA) flow diagram.
3.The figures in the article does not directly show the development trend of deep learning application in prostate cancer. I would suggest you to add some figures like “Trends in frequency ofmethods used or other predictors within the reviewed articles of this study.” PMID:31478869
4.The manuscript lack of literature for in-depth analysisand the topic of this study was “Deep Learning”, then the author might describe the detailed of supervised algorithm.
Author Response
In the manuscript “Role of Deep Learning in Prostate Cancer management: Past, present, and future based on a comprehensive literature review” by Nithesh et al, the authors have done a study on the present applications of Deep Learning (DL) in prostate cancer diagnosis and treatment. It’s a good job that has certain suggestive significance for the rapid development of artificial intelligence in urinary tumors. I have several suggestions to improve the manuscript:
1. Why the database here not used in your study? PsycINFO, Econlit, CINAHL, EMBASE
Response: Thank you for making us aware of the above-mentioned database libraries. We performed a thorough search through the database engines mentioned in the manuscript and as per your advice performed a search in the above-mentioned database libraries as well, but no additional article/ study was found.
2. I would suggest you display article selection by Preferred Reporting Items for Systematic Reviews and Meta-Analyses (PRISMA) flow diagram.
Response: Thank you for the genuine query and comment. This was a narrative review, and although we adhered closely to PRISMA guidelines, our search strategy did not strictly adhere to the PRISMA guidelines so we do not have the PRISMA flowchart.
3. The figures in the article do not directly show the development trend of deep learning application in prostate cancer. I would suggest you add some figures like “Trends in the frequency of methods used or other predictors within the reviewed articles of this study.” PMID:31478869
Response: As per your advice the necessary figures are added to the main manuscript (Fig 2).
4. The manuscript lack literature for in-depth analysis and the topic of this study was “Deep Learning”, then the author might describe the detailed supervised algorithm.
Response: Thank you for the advice. Information on supervised learning is added to the main manuscript in the form of a figure.
What is supervised learning?
The use of labeled datasets distinguishes supervised learning from other forms of machine learning. Using these datasets, algorithms can be trained to better classify data or predict results. The model's accuracy can be measured and improved over time using labelled inputs and outputs.
Based on data mining, supervised learning can be categorized into two types: classification and regression: (a) Classification tasks rely on an algorithm to reliably assign test data to specified groups. For example Supervised learning algorithms can differentiate spam from the rest of the incoming emails. Classification methods include linear classifiers, support vector machines, decision trees, and random forests. (b) Regression is used to learn about the relationship between dependent and independent variables. Predicting numerical values based on various data points is possible with regression models. Linear regression, logistic regression, and polynomial regression are all common regression algorithms.
What is unsupervised learning?
For the analysis and clustering of unlabeled data sets, unsupervised learning makes use of machine learning methods. These algorithms, which are referred to as "unsupervised," find hidden patterns in data without the aid of a human intervention. Three key tasks are performed by unsupervised learning models: (a) clustering, (b) association, and (c) dimensionality reduction.
Using data mining techniques such as clustering, it is possible to create groups of unlabeled data that are similar or dissimilar. Similar data points are grouped together according to the K value in K-means clustering algorithms. This method is useful for a variety of things, including image segmentation and image compression. Another unsupervised learning technique is an association, which employs a separate set of criteria to discover connections among the variables in a dataset.
Dimensionality reduction is a learning approach used when the number of features (or dimensions) in a dataset is too large. It minimizes the quantity of data inputs while yet maintaining the integrity of the data. To enhance image quality, auto encoders often utilize this technique to remove noise from visual data before it is processed further.
Over the last decade, imaging technology has significantly improved, which has made it easier for us to apply computer vision technologies for the classification and detection of diseases [3]. With the advancements in graphics processing units (GPUs) and their computational power to perform parallel processing, computer vision processing is more accessible today. Deep learning is also being used for data management, chatbots, and other facilities that aid in medical practice. NLP practices used in finding patterns in multimodal data have been shown to increase the learning system's accuracy of diagnosis, prediction, and performance [4]. However, identifying essential clinical elements and establishing relations has been difficult as these records are usually unordered and disorganized. Urology has been at the forefront of accepting newer technologies to achieve better patient outcomes. This comprehensive review aims to give an insight into the applications of deep learning in Urology.

Round 2
Reviewer 1 Report
The authors responded appropriately to the queries.
Reviewer 2 Report
Thanks to the author for his work and reply. Although no corresponding literature was found for the first question, these databases should also be included in the original text.